# Ultrasound-Assisted Wound (UAW) Debridement in the Treatment of Diabetic Foot Ulcer: A Systematic Review and Meta-Analysis

**DOI:** 10.3390/jcm11071911

**Published:** 2022-03-30

**Authors:** Sebastián Flores-Escobar, Francisco Javier Álvaro-Afonso, Yolanda García-Álvarez, Mateo López-Moral, José Luis Lázaro-Martínez, Esther García-Morales

**Affiliations:** 1Diabetic Foot Unit, Clínica Universitaria de Podología, Facultad de Enfermería, Fisioterapia y Podología, Universidad Complutense de Madrid, 28040 Madrid, Spain; jhflores@ucm.es (S.F.-E.); ygarci01@ucm.es (Y.G.-Á.); matlopez@ucm.es (M.L.-M.); diabetes@ucm.es (J.L.L.-M.); eagarcia@ucm.es (E.G.-M.); 2Instituto de Investigación Sanitaria del Hospital Clínico San Carlos (IdISSC), 28040 Madrid, Spain

**Keywords:** ultrasound assisted wound debridement, diabetic foot ulcers, diabetic foot, treatment

## Abstract

A systematic review and meta-analysis were carried out to investigate the effect of ultrasound-assisted wound (UAW) debridement in patients with diabetic foot ulcers (DFUs). All selected studies were evaluated using the Cochrane risk of bias tool to assess the risk of bias for randomized controlled trials. PubMed and Web of Science were searched in October 2021 to find randomized clinical trials (RCT) assessing the effect of UAW debridement on DFUs. RevMan v5.4. was used to analyze the data with the Mantel–Haenszel method for dichotomous outcomes. A total of 8 RCT met our inclusion criteria, with 263 participants. Concerning the healing rate comparing UAW versus the control group, a meta-analysis estimated the pooled OR at 2.22 (95% CI 0.96–5.11, *p* = 0.06), favoring UAW debridement, with low heterogeneity (x^2^ = 7.47, df = 5, *p* = 0.19, I^2^ = 33%). Time to healing was similar in both groups: UAW group (14.25 ± 10.10 weeks) versus the control group (13.38 ± 1.99 weeks, *p* = 0.87). Wound area reduction was greater in the UAW debridement group (74.58% ± 19.21%) than in the control group (56.86% ± 25.09%), although no significant differences were observed between them (*p =* 0.24). UAW debridement showed higher healing rates, a greater percentage of wound area reduction, and similar healing times when compared with placebo (sham device) and standard of care in patients with DFUs, although no statistically significant differences were observed between groups.

## 1. Introduction

Among complications caused by diabetes mellitus, diabetic foot ulcer (DFU) is one of the most serious and costly [1]. Diabetic foot syndrome is defined as the presence of infection, ulceration, or destruction of foot tissues associated with peripheral arterial disease (PAD) and neuropathy [2]. Approximately 19–34% of diabetic patients will develop a DFU during their lifetime, leading to amputation of the affected limb [3,4]. Eighty-five percent of amputations in patients with diabetes will be preceded by the presence of a foot ulcer, reaching a mortality rate of seventy percent at five years after initial amputation [3,5].

Standard of care (SOC) in patients with DFU is based on infection control, use of pressure off-loading devices, PAD management, local wound care, metabolic control of diabetes, and treatment of co-morbidities [6]. Wound debridement is a fundamental part of the local treatment of ulcers and consists of removing devitalized tissue from the wound bed to obtain viable tissue to promote healing [7]. There are different types of debridement, including mechanical, sharp/surgical, autolytic, enzymatic, or biological debridement [8]. The International Working Group on the Diabetic Foot (IWGDF) recommends sharp/surgical debridement in preference to other techniques because it is the least expensive, fastest method of wound bed preparation and is available in all geographic areas [7,9]. Sharp/surgical debridement requires specific clinical skills as there is the potential for extensive damage to the wound bed with exposure of bone, joint tissue, or ligament [7].

Currently, in developed countries, it is estimated that approximately 50% of patients with diabetes and foot ulceration have PAD, and it is estimated that 65% of DFUs have an ischemic component; therefore, an effective alternative to traditional debridement techniques is ultrasound-assisted wound (UAW) debridement, which is useful when sharp/surgical debridement is contraindicated, such as in patients with poor vascular status [8,10,11]. 

There are two modalities of UAW debridement—contact and non-contact—which have identical effects on wound healing. The only difference between the two modalities is how ultrasound is applied: non-contact UAW delivers ultrasound energy to the wound bed through a fine mist of sterile saline applied at a distance between 5 and 15 mm from the wound [12,13].

The effectiveness of UAW debridement is due to the cavitation and micro-streaming effects of ultrasound. Cavitation refers to the formation of oscillating gas microbubbles in a fluid medium; when it occurs, microbubbles expand, contract, and implode, allowing the removal of non-viable tissue and biofilms without damaging healthy tissue [14,15,16]. Likewise, micro-streaming refers to the flow of interstitial fluids caused as a result of the vibration generated by the ultrasound device; this effect alters cell membrane permeability and second messenger activity, resulting in increased protein synthesis, mast cell degranulation, and increased growth factor production, which ultimately leads to neo-angiogenesis and fibroblast stimulation at the wound site [17,18].

Several studies have shown that UAW treatment favors granulation tissue formation in the wound bed, resulting in increased healing rates and reduced healing times of hard-to-heal wounds [12,19,20]. A case series published by Lázaro-Martinez et al. on the effect of UAW debridement in neuroischaemic DFUs showed a significant bacterial load reduction, independent of bacterial species. Bacterial load reduction was associated with improved clinical wound characteristics and a significant reduction in wound size [21]. A recent open-label randomized and controlled parallel clinical trial comparing UAW debridement versus surgical debridement in patients with DFU over a 6-week treatment period demonstrated a significant improvement in cell proliferation and reduction of bacterial load, resulting in a reduction in healing time with the use of UAW debridement [22].

To build upon these previous findings, the purpose of this systematic review and meta-analysis is to assess the effect of UAW debridement on cure rates, time to healing, and wound area reduction in patients with DFUs.

## 2. Material and Methods

This systematic review and meta-analyses have been performed following the general guidelines and recommendations of preferred reporting items for systematic reviews and meta-analyses (PRISMA) [23].

### 2.1. Literature Search

The PubMed and Web of Science databases were systematically searched in October 2021. The keywords used for the search were: (((ultrasound) OR (ultrasonic)) AND (debridement)) AND (diabetic foot ulcer). To identify additional reports, the reference list of retrieved studies was cross-checked.

### 2.2. Article Selection

Inclusion criteria were randomized controlled clinical trials (RCTs) published in English, including humans >18 years old and assessing the effects of UAW debridement compared to SOC and placebo in DFUs. Exclusion criteria were animal or in vitro studies, studies on the wound of different etiologies, and studies with insufficient data for analysis. 

Title and abstract review were performed independently by two reviewers (S.F.-E. and F.J.Á.-A.); any discrepancies between the two reviewers were discussed with a third reviewer (J.L.L.-M.).

The articles included in the systematic review were divided into two groups, one comparing UAW debridement versus placebo and the other UAW debridement versus SOC. Placebo refers to the use of a sham device, whereas SOC is based on local wound care using moist dressings, infection control, and use of pressure off-loading devices [6].

### 2.3. Data Collection

A customized Microsoft Excel spreadsheet was used to extract the data from the studies. The extracted data included: author name, year of publication, study design, number of included patients, intervention evaluated and comparison, and outcome measures (healing rate, time to healing, and wound area reduction). 

### 2.4. Assessment of Risk of Bias and Quality of Evidence

Risk of bias in each of the included studies was estimated using the Cochrane risk of bias tool [24], according to six specific domains: random-sequence generation (selection bias), allocation concealment (selection bias), blinding (performance bias and detection bias), incomplete outcome data (attrition bias), selective reporting (reporting bias), and other bias (including supposed financial support). Each domain was evaluated for low, high, or unclear risk for bias. Further, the quality of the evidence was judged to be high, moderate, low, or very low according to the grading of recommendations, assessment, development, and evaluations (GRADE) system, based on the risk of bias, inconsistency, indirectness, imprecision, and publication bias (GRADEpro/GDT, https://gdt.gradepro.org/ accessed on 15 March 2022) [25].

The assessment was conducted independently by two reviewers (S.F.-E. and F.J.Á.-A.); any discrepancies between the two reviewers were discussed with a third reviewer (J.L.L.-M.).

### 2.5. Statistical Analysis

Frequency and descriptive analyses were performed using SPSS (IBM Corp. Released 2017. IBM SPSS Statistics for Macintosh, Version 25.0. Armonk, NY, USA: IBM Corp.). 

The Shapiro–Wilk test was used to verify the assumption of normality of all continuous variables. Student’s *t*-test and Mann–Whitney U test were performed for normally and abnormally distributed quantitative variables, respectively.

The patient was the unit of analysis for all studies. When studies comparing similar interventions reported the same outcome measures, their data were combined for meta-analysis. Review Manager (RevMan, Version 5.4. The Cochrane Collaboration, London, UK, 2020) was used to analyze the data with the Mantel–Haenszel method for dichotomous outcomes and inverse variance method for continuous outcomes according to a fixed-effect or random-effects model. Estimates of the intervention’s effects are expressed as the odds ratio (OR) (95% CI) for dichotomous outcomes and standardized mean difference (SMD) (95% CI) for continuous outcomes. 

Heterogeneity was estimated clinically and methodologically, and when I square (I^2^) exceeded 50%, a random-effects model was used [26]. The significance of any discrepancies in the estimates of the treatment effects from the different trials was assessed using the Cochrane test for heterogeneity and the I^2^ statistic.

## 3. Results

### 3.1. Literature Search

A total of 155 manuscripts were identified from the literature. After screening the titles and abstracts, we identified 126 potential records. After a full-text review, a total of eight RCTs met the selection criteria and were included in this systematic review [20,22,27,28,29,30,31,32] (Figure 1).

### 3.2. Assessment of Risk of Bias and Quality of Evidence

The studies included in the systematic review were published between 2005 and 2020 and included 263 participants. The sample size ranged from 8 to 60 patients per study, with a mean size of 32.87 ± 21.08 patients. 

According to the GRADE system, quality of the evidence was considered “very low” because of the imperfect study design, small sample size, significant heterogeneity, and potential publication bias. The results are summarized in Table 1.

The risk of bias assessment of the eight RCTs included in the systematic review is summarized in Figure 2 and Figure 3.

### 3.3. Outcome Measures

The number of patients included in each study, type of intervention, rate, time to healing, and percentage reduction in wound area reduction are shown in Table 2. According to intervention, UAW debridement was compared with placebo (sham device) and SOC in three [27,29,32] and five trials, respectively [20,22,28,30,31]. Placebo and SOC refer to the control group.

#### 3.3.1. Healing Rate

A total of 6 studies, including 226 patients, compared the effects of UAW debridement in relation to healing rate versus control group [20,22,27,29,31,32]. A meta-analysis of this data estimated the pooled OR at 2.22 (95% CI 0.96–5.11, *p* = 0,06), favoring UAW debridement, with low heterogeneity (x^2^ = 7.47, df = 5, *p* = 0.19, I^2^ = 33%), although no statistically significant differences were observed between groups (Figure 4).

#### 3.3.2. Time to Healing

A total of 4 studies, including 158 patients, provided data about the healing times of DFUs and compared the effect of UAW debridement versus the control group [20,22,27,31]. Time to healing was similar in both groups, and no statistically significant differences were observed; 14.25 ± 10.10 weeks in the UAW debridement group versus 13.38 ± 1.99 weeks in the control group (*p* = 0.87). 

#### 3.3.3. Wound Area Reduction

A total of 5 studies, including 186 patients, compared the effects of UAW debridement in wound area reduction versus the control group [20,22,32]. Wound area reduction was greater in the UAW debridement group (74.58 ± 19.21%) than in the control group (56.86 ± 25.09%), although no significant differences were observed between them (*p* = 0.24).

## 4. Discussion

This systematic review with meta-analysis shows that UAW debridement in patients with DFUs is associated with higher healing rates, a greater percentage of wound area reduction than placebo and SOC, and similar healing times between UAW debridement and control groups.

In the clinical trials included in this systematic review, UAW debridement was conducted using a low-frequency ultrasound device; the frequencies used ranged from 22 to 60 kHz. There are two modalities of UAW debridement: contact and non-contact. Both are based on the effect of cavitation and micro-streaming to remove non-viable tissue from the wound bed. As the name suggests, non-contact UAW debridement generates the same effect but with a lower intensity and without direct contact with the wound surface [31].

Although the healing rates favored the UAW group with OR at 2.22 (95% CI 0.96, 5.11), no statistically significant differences were observed concerning the control group (placebo and SOC). These results could be a consequence of the small sample sizes observed in the different included studies, which ranged from 8 to 60 patients with DFUs.

The effect of UAW debridement was compared with placebo (sham device) in three studies [27,29,32]. The follow-up time of the studies ranged from 4 to 12 weeks, the frequency of debridement application varied from 1 to 3 times per week, and DFUs included were classified as Wagner 1, 2, and 3. Application time of UAW debridement was only reported in the studies published by Ennis et al. [27] and Rastogi et al. [32], and were 4 min/cm^2^ and 15 min/cm^2^, respectively. In all studies compared to placebo, the rate of DFU healing was higher for the UAW debridement group, with values of 23.5–100% versus 11.5–25%. 

SOC effect compared to UAW debridement on DFUs was reported in five studies [20,22,28,30,31]. The follow-up time of the studies ranged from 5 to 24 weeks, the frequency of debridement application varied from 1 to 3 times per week, and DFUs included were classified according to Wagner [33] and Texas [34] classifications. Only two of five studies analyzed reported on application time of UAW debridement; in the study conducted by Lázaro-Martínez et al. [22], only neuroischaemic DFUs were included, and application time of UAW debridement was 2–3 min/cm^2^, whereas the RCT published by Amini et al. [20] included neuropathic and neuroischaemic DFUs and application time of UAW debridement was 1 min/cm^2^.

Regarding studies comparing UAW debridement with SOC, three studies reported on the healing rate. Amini et al. [20] and Lázaro-Martínez et al. [22] showed that the healing rate was higher with UAW debridement than with SOC; 60% and 85.1%, respectively. In contrast, Michailidis et al. [31] found a higher healing rate in the SOC group than in the UAW debridement group (83.3% versus 62.5%).

In terms of healing time, UAW debridement appears to have similar healing times to the control group. These findings could be caused by the variability of DFUs included in the RCTs, as healing time will differ depending on the wound depth and presence or absence of infection or ischemia. Another factor to consider is the variability of DFUs classification systems used in the RCTs (Wagner [33] and Texas [34] classifications).

Healing time in studies compared to placebo was only reported in the study published by Ennis et al. [27], being shorter in the UAW debridement group than the placebo group (9.12 ± 0.58 versus 11.74 ± 0.22 weeks). In relation to healing time of DFUs in studies comparing UAW debridement versus SOC, Amini et al. [20] and Lázaro-Martínez et al. [22] showed that healing times were shorter with UAW debridement (8.8 ± 12 and 9.7 ± 3.8 weeks) than with SOC (11.6 ± 11.2 and 14.8 ± 12 weeks). Michailidis et al. [31] found that the time to healing was greater in the UAW debridement group than in the SOC group (29.4 ± 10.07 and 15.4 ± 6.1 weeks).

The results obtained in relation to healing rate and healing time in the study carried out by Michailidis et al. [31] in favor of the SOC group could be related to the small sample size and with an application time of UAW debridement, which was not precisely determined.

In addition, the reduction of wound area was greater in patients with DFUs where UAW debridement was applied. The absence of statistically significant results can be explained by the existence of the wide variation in the application time for UAW debridement and the frequency of debridement treatments, ranging from once per week to three times per week. Regarding the application time of UAW debridement, authors such as Amini et al. [20] established in their study an application time of 1 min/cm^2^, whereas in the study by Bajpai et al. [29], the application time was 15 min/cm^2^. The great difference in application times and frequency of UAW debridement is due to the use of ultrasound devices with different modalities (contact or non-contact ultrasound devices).

The percentage of wound area reduction in studies compared to SOC was referenced in four studies, in all of which wound area reduction was greater in the UAW debridement group than in the SOC group, with values of 43–87.9% versus 24.4–82.4%. Likewise, the percentage of wound area reduction in studies compared to placebo was referenced in one study [32]; this outcome showed a greater wound area reduction in the UAW debridement group (69.4 ± 23.2%) than the placebo group (59.6 ± 24.9%).

Regarding the level of evidence and the degree of recommendation of the included studies, all were controlled and randomized clinical trials, with a level of evidence 1b and degree of recommendation A. In 2/8 and 4/8 of the included studies, there were a high risk of bias in the blinding of results and of participants and/or professionals, respectively. There was a medium risk of bias in the allocation concealment in 7/8 studies and a low risk of bias in the random sequence in 7/8 studies. In general, in all studies, there was an unclear or low risk of bias in some of the items, mainly due to lack of information. Despite all studies being randomized clinical trials, the high risk of bias in the blinding of patients and professionals, together with the lack of information in most of the studies, limits the conclusions of this review with meta-analysis. The great difficulty in blinding patients and professionals when applying this type of instrumentalized technique should be emphasized.

To our knowledge, this study is the first systematic review with meta-analysis to assess the effect of UAW debridement on healing rates, time to healing, and wound area reduction in patients with DFUs. Therefore, it is not possible to establish comparisons with other similar previous studies.

A factor to consider regarding the literature search is the restriction of the included publications to English. The main limitation of this systematic review with meta-analysis is the small sample size of the RCTs included, which limits the generalizability of the results. Another important limitation is heterogeneity between the different RCTs, in terms of the clinical characteristics of the DFUs included (depth, infection, or ischemia), study follow-up time, time of application, and frequency of application associated with the type of ultrasound used (contact or non-contact). Finally, the lack of certain information in the studies is another limitation in evaluating some variables since it prevents the inclusion of some studies in our meta-analysis. Having this data could increase the information provided by this systematic review with meta-analysis. 

Further clinical trials with low risk of bias, using control groups, with clear randomization and blinding of results could help clarify our conclusions. In addition, it is recommended to calculate the sample size of each treatment group and standardize the follow-up period of the study, the clinical characteristics of the DFUs included, and to establish a protocol about application time and frequency of UAW debridement.

## 5. Conclusions

Compared with placebo (sham device) and SOC, UAW debridement shows higher healing rates, a greater percentage of wound area reduction, and similar healing times in patients with DFUs, but greater quality evidence is needed to confirm these findings. UAW debridement could be an effective alternative when traditional debridement techniques are not available or are contraindicated for use. Limitations of this systematic review with meta-analysis include the small sample sizes and wide heterogeneity among RCTs in terms of clinical characteristics of DFUs, study follow-up time, application time and application frequency associated with the type of ultrasound used.

## Figures and Tables

**Figure 1 jcm-11-01911-f001:**
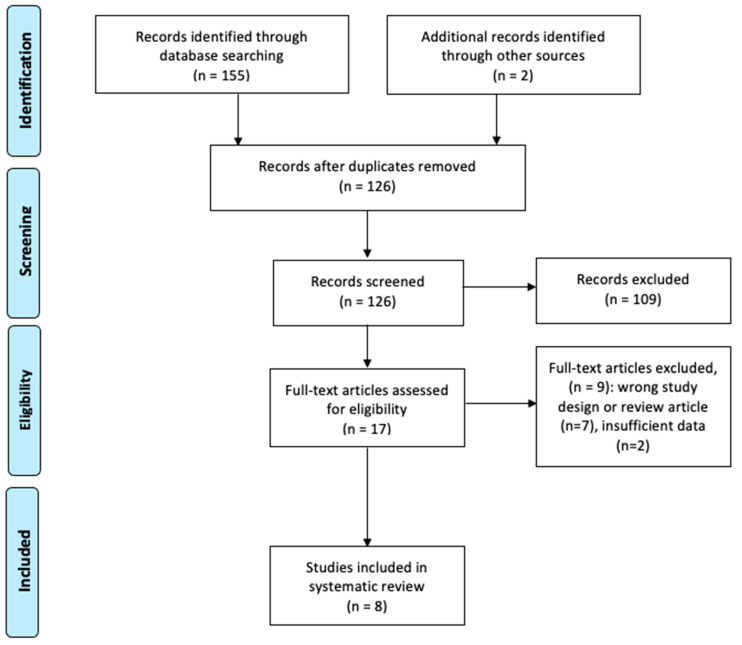
Flow diagram of the literature search and study selection for the systematic review.

**Figure 2 jcm-11-01911-f002:**
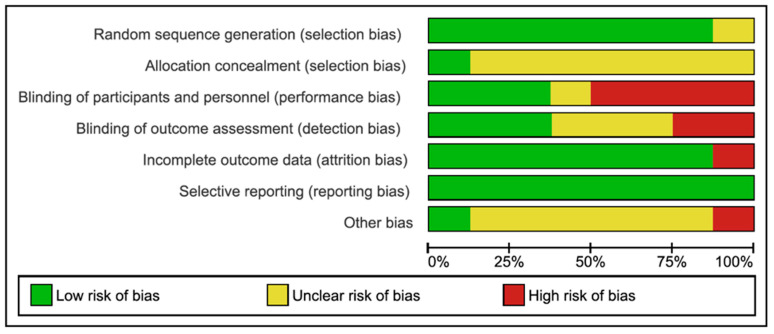
Risk of bias graph: Review authors’ judgments about each risk of bias item presented as percentages across all included studies.

**Figure 3 jcm-11-01911-f003:**
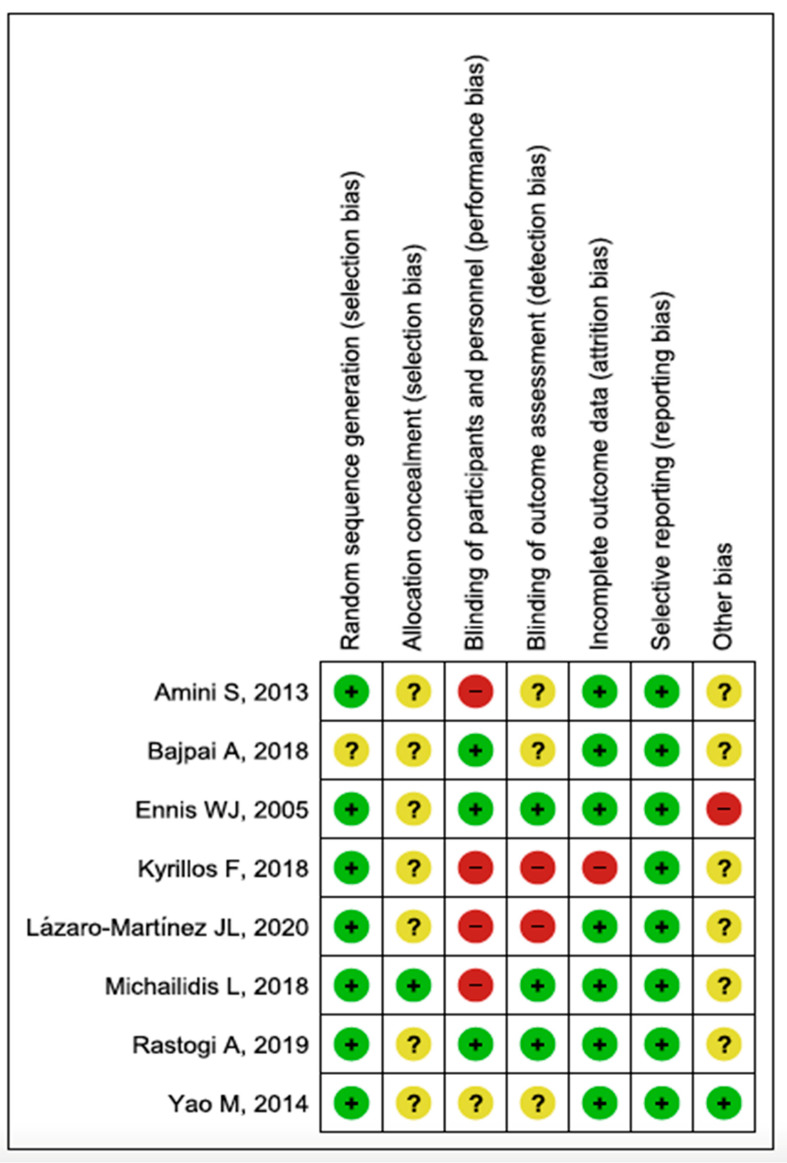
Risk of bias summary: Review authors’ judgments about each risk of bias item for each included study. Green: Low risk of bias. Yellow: Unclear risk of bias. Red: High risk of bias.

**Figure 4 jcm-11-01911-f004:**
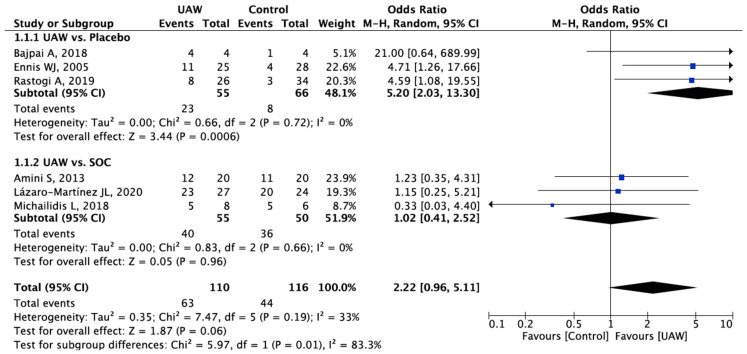
Forest plot of UAW debridement versus control (placebo and SOC) for complete healing rate. Bold text means overall outcomes per subgroups. Blue square: Odd Ratio for each study (measure of effect of each study).

**Table 1 jcm-11-01911-t001:** GRADE assessment for the effect of UAW on the healing of DFU.

Outcome	Number of Studies	Study Design	Certainty Assessment						
			Risk of Bias	Inconsistency	Indirectness	Imprecision	Other Considerations	Effect	Certainty
Healing Rate	6	RCT	Serious ^a^	Serious ^b^	Not serious ^c^	Serious ^d^	Publication bias strongly suspected	OR (95% CI) 2.22 (0.96, 5.11)	⨁◯◯◯◯◯ Very low
Time to Healing	3	RCT	Serious ^a^	Serious ^b^	Not serious ^c^	Serious ^d^	Publication bias strongly suspected	SMD (95% CI) −1.41 (−3.43, 0.61)	⨁◯◯◯◯◯◯ Very low
Wound Area Reduction	3	RCT	Serious ^a^	Not serious	Not serious ^c^	Serious ^d^	Publication bias strongly suspected	SMD (95% CI) 0.23 (−0.09, 0.55)	⨁◯◯◯◯◯◯ Very low

^a^ The randomization method, allocation concealment, and blinding method of some included studies were not clear, and some studies did not carry out blinding method. ^b^ Differences were observed between the studies in relation to the time of application and frequency of the intervention as well as in the follow-up time of the patients (high heterogeneity). ^c^ All included studies were related to research questions and no indirect comparisons were made. ^d^ The sample size was small. ⨁ and ◯ means very low certainly.

**Table 2 jcm-11-01911-t002:** Characteristics of the RCTs included in the systematic review.

Author/Year	Number of Participants	Intervention	Healing Rate (%)	Time to Healing (Weeks)	Wound Area Reduction (%)
Ennis [27]/2005	Arm 1: 25 Arm 2: 28 Total: 55	Arm 1: UAW Arm 2: Placebo	Arm 1: 11 (40.7%) Arm 2: 4 (14.3%)	Arm 1: 9.12 ± 0.58 w Arm 2: 11.74 ± 0.22 w	–
Amini [20]/2013	Arm 1: 20 Arm 2: 20 Total: 40	Arm 1: UAW Arm 2: SOC	Arm 1: 12 (60%) Arm 2: 11 (55%)	Arm 1: 8.8 ± 12 w Arm 2: 11.6 ± 11.2 w	Arm 1: 87.9 ± 33.8% Arm 2: 82.4 ± 33%
Yao [28]/2014	Arm 1: 4 Arm 2: 4 Arm 3: 4 Total: 12	Arm 1: UAW 3/w Arm 2: UAW 1/w Arm 3: SOC	–	–	Arm 1: 86% Arm 2: 25% Arm 3: 39%
Bajpai [29]/2018	Arm 1: 4 Arm 2: 4 Total: 8	Arm 1: UAW Arm 2: Placebo	Arm 1: 4 (100%) Arm 2: 1 (25%)	–	–
Kyrillos [30]/2018	Arm 1: 12 Arm 2: 11 Total: 23	Arm 1: UAW Arm 2: SOC	–	–	Arm 1: 43% Arm 2: 24.4%
Michailidis [31]/2018	Arm 1: 8 Arm 2: 6 Total: 14	Arm 1: UAW Arm 2: SOC	Arm 1: 5 (62.5%) Arm 2: 5 (83.3%)	Arm 1: 29.4 ± 10.07 w Arm 2: 15.4 ± 6.1 w	–
Rastogi [32]/2019	Arm 1: 26 Arm 2: 34 Total: 60	Arm 1: UAW Arm 2: Placebo	Arm 1: 8 (23.5%) Arm 2: 3 (11.5%)	–	Arm 1: 69.4 ± 23.2% Arm 2: 59.6 ± 24.9%
Lázaro-Martínez [22]/2020	Arm 1: 27 Arm 2: 24 Total: 51	Arm 1: UAW Arm 2: SOC	Arm 1: 23 (85.1%) Arm 2: 20 (83.3%)	Arm 1: 9.7 ± 3.8 w Arm 2: 14.8 ± 12 w	Arm 1: 86.6 ± 83.8% Arm 2: 78.94 ± 68.6%

## Data Availability

Data sharing does not apply to this article as no datasets were created during the current study.

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
