# Peer review of "Ultrasound-Assisted Wound (UAW) Debridement in the Treatment of Diabetic Foot Ulcer: A Systematic Review and Meta-Analysis"

_jcm, 2022, doi:10.3390/jcm11071911_

Round 1
Reviewer 1 Report
Dear authors,
Congratulations on the work. However, the execution and review methods description have some issues to solve/clarify :
1) Search strategy: indexed terms (MeSH) were incorporated into the search strategy? What is the data of the search strategy?
2) Restriction of language and time of publication. Why are the authors restricted to English? Was there a restriction on the time of publication?
3) Is there a protocol of review published? If not, why?
4) Data extraction was performed by whom? Were revised for someone?
5) What about missing data?
6) Was planned to perform a subgroup analysis? If not, why?
7) Was planned to perform a sensitivity analysis? If not, why?
8) Although described as 6 domains, the authors described 7 domains of risk of bias: blinding is considered as 2 domains
9) Decision of meta-analysis. As described by the author, there is essential clinical heterogeneity among the included studies. Therefore, a meta-analysis including different studies was not indicated; "mixing apples with oranges". What are the authors thinking about this issue?
10) In line 276: the expression "medium risk of bias" is not a term impregnated. The tool admits unclear, low or high risk of bias.
11) The authors do not indicate the level of certainty about the evidence. It is essential the analysis the factors that downgrade and upgrade the evidence incorporating the risk of bias, sample size, inconsistency, heterogeneity etc, in each outcome researched. Use the GRADE system.
Best regards,
Reviewer 2 Report
This paper aims to reviewing, all published RCTs about performance rates of ultrasound-assisted wound (UAW) debridement in diabetic foot ulcers. After excluding papers non eligible for analysis only eight studies were considered. Comparison of UAW were SOC or placebo, both considered as control groups. Healing rates were absolutely identical when comparing UAW and SOC , The advantage of UAW was instead with an OR=2.22 when considering both SOC and placebo as controls, but also in this case the CIS (0.96-5.55) didn’t permit any full significance. A first point: one should better state what was placebo and what was SOC. Times to healing and wound area reductions were non significantly different between groups. In conclusion no clear superiority can be determined especially when comparing UAW and SOC in OR of healing. In addition to all limitations associated with this meta-analysis clearly stated in Discussion all this should be clearly discussed and stated in Conclusions, without giving non exact impressions as from the actual Abstract and Discussion of the paper. The last phrase of the Abstract, indeed, is in this case non exact regarding healing rates in comparison to SOC. This doesn’t exclude that that information from this paper may be, if correctly given, useful to all scientific community as well as to clinicians involved in the care of DF ulcers.
A further point: from means and SD of both times to healing and wound areas it seems that they aren’t normally distributed: may be giving values as medians (IQR) could be more exact.
Round 2
Reviewer 2 Report
Revisions improve enough this paper. Only a minor point: please add p valuese to ORs in Figure 4 (UAW vs placebo and UAW vs SOC and claerly state that, although closde to significance, UAW vs SOC is not fully significant (95%CIs contain 1).
